# Deep Supervised Attention Network for Dynamic Scene Deblurring

**DOI:** 10.3390/s25061896

**Published:** 2025-03-18

**Authors:** Seok-Woo Jang, Limin Yan, Gye-Young Kim

**Affiliations:** 1Department of Software, Anyang University, 22, 37-Beongil, Samdeok-ro, Manan-gu, Anyang 14028, Republic of Korea; swjang7285@gmail.com; 2School of Software, Soongsil University, 369, Sangdo-ro, Dongjak-gu, Seoul 06978, Republic of Korea; yanlimin@naver.com

**Keywords:** dynamic deblurring, multiple loss function, multi-scale network, supervised attention, recurrent network, feature mapping

## Abstract

In this study, we propose a dynamic scene deblurring approach using a deep supervised attention network. While existing deep learning-based deblurring methods have significantly outperformed traditional techniques, several challenges remain: (1) Invariant weights: Small conventional neural network (CNN) models struggle to address the spatially variant nature of dynamic scene deblurring, making it difficult to capture the necessary information. A more effective architecture is needed to better extract valuable features. (2) Limitations of standard datasets: Current datasets often suffer from low data volume, unclear ground truth (GT) images, and a single blur scale, which hinders performance. To address these challenges, we propose a multi-scale, end-to-end recurrent network that utilizes supervised attention to recover sharp images. The supervised attention mechanism focuses the model on features most relevant to ambiguous information as data are passed between networks at difference scales. Additionally, we introduce new loss functions to overcome the limitations of the peak signal-to-noise ratio (PSNR) estimation metric. By incorporating a fast Fourier transform (FFT), our method maps features into frequency space, aiding in the recovery of lost high-frequency details. Experimental results demonstrate that our model outperforms previous methods in both quantitative and qualitative evaluations, producing higher-quality deblurring results.

## 1. Introduction

Traditional image deblurring algorithms reconstruct images based on a specific image model, and the image degradation process can be expressed as follows:
(1)B=K ∗ S+n,
where B, S, n, and K represent the blurred image, latent sharp image, noise, and unknown blur kernel, respectively. To reconstruct the latent image S, a deblurring algorithm must accurately estimate the blur kernel K and then perform a deconvolution operation on the blurred image using K to recover the sharp image. However, different combinations of sharp images and blur kernels can produce the same blurred image after convolution, making the deblurring problem inherently ill-posed.

Kim et al. [1] introduced the concept of dynamic scene deblurring, highlighting that blur is caused by various factors, such as camera shake and object motion, leading to non-uniform blur in dynamic scenes. Despite this, traditional methods that rely on prior knowledge to estimate the blur kernel often overlook the non-uniform nature of blur, making accurate kernel estimation unrealistic and prone to artifacts. Additionally, most traditional image restoration methods based on prior knowledge use iterative optimization, which involves tuning a large number of parameters and significantly increases computational overhead. Consequently, the performance of traditional deblurring methods could be improved. With the advent of deep learning, dynamic image deblurring has made significant strides in both performance and efficiency. Deep learning-based methods use multiple image pairs to create mapping functions from distorted to sharp images, eliminating the need to estimate complex prior information and reducing errors associated with traditional methods. Sun [2] and Schuler [3] were the first to introduce convolutional neural networks (CNNs) for image deblurring, where CNNs still estimate the blur kernel. Although CNN-based approaches can improve deblurring performance by more accurately predicting the blur kernel, they remain limited to specific types of blur and struggle with spatially varying blurs. To address these limitations, Nah et al. [4] proposed a multi-scale CNN-based image restoration approach that directly recovers latent images without assuming a specific blur kernel model, thereby avoiding artifacts caused by kernel estimation errors. However, their method does not account for the temporal information of blurred images, which is a critical aspect given that blur features span both temporal and spatial dimensions.

Building on the work of Nah et al., Zhang et al. [4,5] introduced an RNN structure to capture temporal information in images. To address the issue of parameter redundancy in multi-scale networks, Xin et al. [6] proposed a multi-scale recurrent network with parameter sharing, significantly reducing network complexity. Subsequently, Kupyn et al. [7] utilized a generative adversarial network (GAN) to input blurred images into the generator and output sharp images, with a discriminator supervising the quality of the generated images. To leverage multi-scale features, they incorporated a feature pyramid network in the generator, combining feature information across different scales to better handle image blurring. Kuldeep et al. [8] further enhanced deblurring by introducing self-attention mechanisms to capture non-local spatial dependencies between features, improving the network’s ability to manage spatial variations. However, they overlooked the significant computational overhead introduced by self-attention.

Recent advancements in deep learning-based deblurring have introduced various approaches utilizing CNNs, RNNs, and self-attention mechanisms to address dynamic scene blurs. However, many of these methods suffer from inherent redundancy in RNN architectures and the high computational cost associated with self-attention modules. To overcome these limitations, various studies are actively being conducted [9,10,11].

The proposed deep supervised attention network (DSANet) integrates an optimized ConvLSTM-based encoder–decoder framework with a novel supervised attention module. This multi-scale, coarse-to-fine network not only streamlines feature propagation across multiple scales by reducing redundancy but also directs computational resources towards the most informative features. As a result, the DSANet addresses the limitations of existing approaches and achieves superior performance in both quantitative and qualitative evaluations. The main contributions of this study include the following:An optimized ConvLSTM-based encoder–decoder structure that accelerates network integration and enhances the learning of the spatial-temporal features.A newly proposed supervised attention module that mitigates the high computational overhead of self-attention by guiding the model to focus on features most relevant to blur information when transferring features between different network scales.The introduction of a multi-loss function based on the fast Fourier transform (FFT), enabling the model to learn deblurring features in the frequency domain.The development of a new dataset collected in diverse environments, which outperforms existing datasets in several aspects, reducing the challenges posed by discrepancies between simulated data and real-world images.

## 2. Related Works

### 2.1. Encoder–Decoder Structure

Image segmentation has seen significant advancements, largely due to the adoption of encoder–decoder architectures. Recently, these systems have been widely used in various computer vision tasks [12,13]. As illustrated in Figure 1, an encoder–decoder network in computer vision refers to a symmetric architecture built with CNNs. It typically consists of convolutional layers, pooling layers, and batch normalization layers. During the encoding stage, the input data are gradually transformed into feature maps with smaller spatial dimensions and an increased number of channels. In the decoding stage, these compressed feature maps are converted back to the input format through deconvolution or up-sampling operations [14,15]. The encoder–decoder architecture is optimized for faster network convergence and improved gradient propagation. However, its application in image restoration has been limited, as batch normalization is highly sensitive to batch size.

### 2.2. Multi-Scale Network

In many computer vision tasks, various forms of coarse-to-fine or multi-scale architectures are commonly employed [16,17]. The core idea behind multi-scale networks is that images at different scales provide varying degrees of feature information. Smaller-scale images are well suited for capturing global semantic information, which can then guide the analysis of larger-scale images with broader receptive fields and more comprehensive global information. However, the requirements for multi-scale structures can vary across different computer vision tasks. While existing multi-scale structures can address the receptive field needs in deblurring tasks, the use of numerous convolutional layers based on residual blocks to capture long-range dependencies significantly increases the model’s complexity.

### 2.3. Self-Attention

Attention mechanisms [18] were initially introduced to enhance the translation performance of neural machine translation (NMT) systems and have recently gained prominence in computer vision. A typical example is the CNN. In CNNs, each convolutional layer focuses on a localized region defined by the kernel size. Although the receptive field expands over time, it primarily captures global feature correlations. Researchers have explored integrating attention mechanisms with computer vision, often using masks to enhance the attention process. These masks help identify salient features in an image by applying newly trained weights to each image, enabling deep neural networks to focus on areas that require attention. Wang et al. [19] introduced a combined attention module within an encoder–decoder framework, which refines the feature map and improves network performance. However, the need to calculate a 3D focus map results in significant computational overhead. Woo et al. [20] proposed the convolutional block attention module (CBAM), a hybrid attention module that extracts meaningful features across two channel dimensions, enhancing adaptive feature learning. This module can be seamlessly integrated into other networks, but attention mechanisms still incur high computational costs. Hu et al. [21] developed the squeeze-and-excitation network (SENet), which computes global image information at the feature channel level. Tsai et al. [22] introduced the blur-aware attention network (BANet), which achieves accurate and efficient deblurring with a single forward pass. The BANet leverages region-based self-attention with multi-kernel strip pooling to address blur patterns of varying magnitudes and orientations and the use of cascaded parallel dilated convolutions to aggregate multi-scale content features.

### 2.4. Residual Block

He et al. [23] introduced the residual block, a highly successful deep learning architecture that enables the training of very deep networks and addresses the issue of vanishing gradients. While the problems of vanishing and exploding gradients have been substantially mitigated by batch normalization and suitable activation functions [24], residual blocks play a crucial role in preventing network degradation. However, traditional residual networks are not ideal for image restoration tasks. To enhance image restoration quality, Wang et al. [25] removed the normalization layer from all residual blocks. In contrast, Nah et al. [4] demonstrated that incorporating batch normalization can improve the performance of deblurring networks. Unlike traditional residual networks, the residual module used in their approach does not include a batch normalization layer.

## 3. Deep Supervised Attention Network (DSANet)

Figure 2 illustrates the overall structure of the proposed DSANet. The network employs a multi-scale recurrent architecture comprising an encoder and a decoder. The encoder extracts features, while the decoder restores the image. A skip connection between the encoder and decoder enhances the receptive field and accelerates the model convergence.

In addition, the network incorporates a blurred attention block that selectively ignores irrelevant features, focusing instead on connectivity features of varying magnitudes to suppress redundant information. Equation (2) shows how the network processes a blurred image to estimate a sharp image.(2)$$R^{i},h^{i}={Net}_{DSANet}(B^{i},R^{(i+1)\prime },h^{(i+1)\prime };\theta_{DSANet}),$$
where *i* represents the scale index, with *i* = 1 indicating the smallest scale; *R^i^* and *h^i^* are the blurred image and estimated latent image at the *i*-th scale, respectively; Net_DSANet_ is a multi-scale recurrent supervised attention network with training parameters denoted as *θ*_DSANet_; the hidden state features, *h^i^*, are propagated across different scales and serve as a dynamic memory that captures essential image structural details as well as blur characteristics, thereby facilitating effective multi-scale feature integration; and (*i* + 1)′ refers to the operation of the adjustment of image or feature size.

When the blurred image is down-sampled to different scales, the sampling coefficient between adjacent scales is 1/2. The process begins by loading the smallest-scale image *R*^3^ into the Encoder1. A down-sampling layer with a stride of 2 reduces the size of the feature map to half its original dimensions and increases the number of channels from 3 to 32 (i.e., [H × W × 3] → [H/2 × W/2 × 32]), where H and W represent the height and width of the feature map, respectively. The number three indicates that the original image has three channels. Four residual blocks, without additional down- or up-sampling, are used to enhance the receptive field of the network and improve image restoration.

The feature map is then passed to Encoder2, which also employs a down-sampling layer with a stride of 2 to reduce the size of the feature map to half of its original size and increase the number of channels from 32 to 64 (i.e., [H/2 × W/2 × 32] → [H/4 × W/4 × 64]). Four residual blocks, without down- or up-sampling, are used to further expand the receptive field and enhance the image restoration effect.

Next, Encoder3 uses a down-sampling layer with a stride of 2 to reduce the feature map size by half and increase the number of channels from 64 to 128 (i.e., [H/4 × W/4 × 64] → [H/8 × W/8 × 128]). Four residual blocks, without down- or up-sampling, are again employed to increase the receptive field and improve image restoration. Subsequently, the ConvLSTM module is introduced to enhance the model’s ability to capture spatial-temporal dependencies.

In the decoder stage, the feature map is input by Decoder1. This stage uses four residual blocks, without down- or up-sampling, to maintain the receptive field and improve image restoration. An up-sampling layer with a stride of 2 is applied to double the size of the feature map and reduce the number of channels from 128 to 64 (i.e., [H/8 × W/8 × 128] → [H/4 × W/4 × 64]).

Decoder2 employs four residual blocks without additional down- or up-sampling to expand the network’s receptive field and enhance image restoration. An up-sampling layer with a stride of 2 is then used to double the size of the feature and reduce the number of channels from 64 to 32 (i.e., [H/4 × W/4 × 64] → [H/2 × W/2 × 32].

Decoder3 also uses four residual blocks without down- or up-sampling to further increase the receptive field and improve image restoration. Another up-sampling layer with a stride of 2 doubles the size of the feature map while keeping the number of channels unchanged (i.e., [H/2 × W/2 × 32] → [H × W × 32]). Finally, in the supervised attention layer, the image is restored to its original image scale [H × W × 3]).

This process can be expressed as follows:(3)$$R^{3},h^{3}={Net}_{DSANet}(B^{3},\theta_{DSANet}),$$
where *R*^3^ and *h*^3^ represent the smallest scale of the restored image and the learned hidden state features, respectively. The supervised attention layer selectively refines *h*^3^ and then concatenates it with the up-sampled features from the previous scale. Similarly, the processing at the next scale is defined as follows:(4)$$R^{2},h^{2}={Net}_{DSANet}(B^{2},R^{(3)\prime },h^{(3)\prime };\theta_{DSANet}),$$
where *R*^2^ and *h*^2^ denote the restored image and learned hidden state features at the second scale, respectively, while R^3′^ and *h*^3′^ represent the up-sampled features from the smallest scale. After processing through the final scale network, the output, which combines the features from the previous scales, is given by the following:(5)$$R^{1},h^{1}={Net}_{DSANet}(B^{1},R^{(2)\prime },h^{(2)\prime };\theta_{DSANet})$$
The proposed network addresses the limitations of previous deblurring networks by incorporating a supervised attention module to achieve enhanced feature extraction capabilities.

### 3.1. Supervised Attenuation Module

Recent work in super resolution has implemented self-attention mechanisms [26], which, while enhancing feature extraction capabilities, have also introduced significant computational overhead.

To improve the restoration network’s ability to perceive blurred features with spatial variations while mitigating computational costs, we propose a blur module based on supervised attention. This module utilizes a supervised attention mechanism to dynamically capture blurred features with spatial variations, making it easier to retrieve a clear latent image. Figure 3 illustrates the supervised attention module.

In the final stage of the multi-scale network, the degraded image *I* is restored using the learned feature map *F_i_^n^*, resulting in the restored image Restored (*x_s_*). The supervised attention module generates an attention feature map using the Sigmoid activation function applied to the recovered image Restored (*x_s_*). The attention feature map’s values ranges from 0 to 1, where higher values indicate greater attention. This attention feature map guides the fusion of features from different scale networks. Since blurring varies spatially, the module’s significance lies in applying different levels of attention to various positions in the blurred image. It follows a selective memory mechanism similar to the forgotten gate in LSTM [27]. Before concatenating features from different scales, unimportant features are selectively forgotten, important features are emphasized, and redundant information is suppressed.

### 3.2. Multi-Loss Function

The rapid advancement in detection and segmentation is largely attributed to effective evaluation metrics. However, a similar robust metric is still needed for low-level vision tasks. The peak signal-to-noise ratio (PSNR) is the primary evaluation metric used in image deblurring. It measures the content loss of an image, with higher PSNR values indicating lower content loss. Despite its widespread use, the PSNR has limitations in accurately assessing image quality. For instance, as shown in Figure 4, an image with the highest PSNR value may not necessarily have the best perceptual quality.

The super-resolution generative adversarial network (SRGAN) [28] addresses this by optimizing loss functions that are more sensitive to human perception. Even if the PSNR value is not exceptionally high, the SRGAN aims to improve the perceptual quality of the restored image. Johnson et al. [29] proposed a multi-loss function based on perceptual loss for style transfer tasks, which improves content representation at the expense of reduced PSNR precision. Later, Jiao et al. [30] and Ignatov et al. [31] introduced auxiliary loss functions to enhance image quality. However, these approaches did not significantly improve perceptual quality compared to single loss function models and often resulted in lower PSNR values.

As with most multi-scale deblurring networks, we initially experimented with a multi-scale single loss function, such as the mean squared error (MSE). However, a single loss function alone did not improve the subjective visual quality of the restored images. Therefore, we developed a new multi-loss function designed to enhance the subjective evaluation of image quality while maintaining or improving the PSNR value. The multi-loss function is defined as follows:(6)$$Loss_{total}=MSE+\mu \times FFT_{MSE},$$
where *FFT_MSE_* uses the Fourier transform to convert the image signal to the frequency domain, and *μ* represents the loss weight. We conducted numerous experiments with different loss weights, such as 0.1, 0.4, and 0.6. The qualitative results demonstrated that a loss weight of *μ* = 0.1 provided the best performance.

The *FFT_MSE_* loss is defined as follows:(7)$$FFT_{MSE}=\frac{1}{M\times N}\sum_{0\leq i\leq N}\sum_{0\leq j\leq M}{\left(FFT(f_{ij})-FFT(f_{ij}^{\prime })\right)}^{2},$$
where *M* and *N* denote the length and width of the image, respectively; *f_ij_* represents the pixel value of the original image; and *f_ij’_* is the pixel value of the target image at position (*i*, *j*). *FFT*(∙) is the fast Fourier transform function. A smaller *FFT_MSE_* value indicates a higher quality of the restored image, while a larger value suggests significant distortion and poor quality. By applying the *FFT*, we can observe features in the frequency domain that are otherwise undetectable. The resulting spectrogram reveals the frequency components of the image, where high-frequency signals correspond to edges and noise, and low-frequency signals correspond to the background. In the frequency domain, we can effectively manipulate high- and low-frequency information for tasks such as image denoising, enhancement, and edge extraction.

## 4. Dataset Construction

The quality of a dataset significantly impacts the advancement of image restoration and the broader field of computer vision research, directly influencing network performance. However, a perfect dataset that includes both degraded real-world images and corresponding ground truth images has not yet been proposed, due to the difficulty in capturing both simultaneously. Most existing datasets rely on synthetic methods [32,33], which often differ significantly from real-world degraded images. To address this issue, Lu et al. [34] and Nimisha et al. [35] explored unsupervised learning methods to mitigate performance problems associated with synthetic datasets. However, their approaches were limited to specific areas such as face and text deblurring. Sun et al. [2] introduced a synthetic dataset comprising 80 natural images and eight blur kernels.

Lai et al. [36] developed a dataset of 100 blurred images collected from real-world scenarios to evaluate deblurring methods. Despite this, none of the available datasets fully address the gap between synthetic and real datasets. Recently, Rim et al. [37] introduced the Realblur dataset, which includes pairs of degraded real-world images and ground truth (GT) images captured simultaneously using an image acquisition system. This system, which uses two cameras to capture blurry and sharp images at different shutter speeds, reduces the discrepancy between synthetic and real datasets. However, there is often a significant offset between the sharp and blurred images, leading to structural differences that make such image pairs unsuitable for training. To address this, Rim et al. [37] suggested photographing only static objects and applying several post-processing steps, including photometric and geometric alignment. Nevertheless, research on deblurring has frequently been conducted in dynamic environments, and a static-only dataset is insufficient for improving deblurring network performance. Additionally, extensive post-processing introduces inefficiencies and high costs. Zhang et al. [38] recently proposed a deblurring network that combines two generative adversarial models [39]: one to learn to blur real images and another to learn to deblur synthetic images. This approach helps mitigate the complexities associated with synthetic and real dataset differences but faces limitations due to the lack of large-scale datasets.

We reviewed the limitations of current synthetic datasets and identified various issues contributing to the gap between synthetic and real datasets. To address these challenges, we propose a new synthetic-based dataset designed to significantly reduce the complexities associated with synthetic data. The new dataset is collected using the method illustrated in Figure 5.

We first record short video clips (1–2 s) of moving objects or scenes under various lighting conditions (daytime, nighttime, indoor, etc.) using a high-speed camera (e.g., 960 fps). This high frame rate ensures that individual frames have minimal motion blur, effectively serving as ground truth sharp images. From the high-speed footage, we designate certain frames as sharp images (e.g., the 19th or 37th frame in the sequence). Because of the short exposure time inherent in high-speed shooting, these frames typically exhibit negligible blur and can be treated as reference ground truth. To synthesize motion blur similar to real-world scenarios, we average a range of consecutive frames around the chosen sharp frame. For instance, we may average the 36 frames or 73 frames centered on the sharp frame, thereby simulating different blur intensities. A larger number of averaged frames results in stronger blur. If necessary, we apply photometric and geometric alignment to correct for any slight camera shifts or changes in lighting conditions. This step helps ensure that the sharp image and its corresponding blurred image align closely, minimizing spatial discrepancies that could adversely affect training. By varying the number of averaged frames, we obtain multiple levels of blur severity from the same scene, all paired with the corresponding sharp ground truth frame. This approach yields a diverse dataset encompassing various motion patterns and blur strengths, which can significantly enhance the robustness and generalization of deblurring models.

## 5. Experimental Results

The quality of the dataset is crucial in image restoration tasks. Generating a dataset for image blur is particularly challenging due to the presence of both uniform and non-uniform blurs, with the latter also known as dynamic blurs. Traditional methods for creating such datasets involve using various blurring kernels to synthesize both uniform and non-uniform blurred images from sharp images. However, this approach often oversimplifies the imaging model, leading to significant discrepancies between synthesized data and real-world images.

Recently, a method involving the averaging of consecutive frames has been proposed to generate blurred images more effectively. Based on this approach, Nah et al. [4] created the GOPRO dataset, which consists of 3214 pairs of sharp and blurred images captured at a resolution of 1280 × 720 pixels. Of these, 2103 pairs were used for training and model development, while the remaining 1111 pairs were reserved for model evaluation. We evaluate the performance of our proposed network by comparing its performance with those of state-of-the-art methods using the GOPRO dataset. Additionally, we assessed the network’s performance both with the GOPRO dataset alone and with the inclusion of our dataset (Figure 6), which was collected using high-speed video capture at 960 fps at a resolution of 1920 × 1080 pixels. Our results demonstrate that the new dataset significantly enhances deblurring performance.

The experiments were implemented using PyTorch and trained on two NVIDIA Titan V GPUs. For each training iteration, randomly cropped 256 × 256-pixel patches were used as input. All training variables were initialized using Kaiming initialization [40]. The Adam optimizer [41] was employed with an initial learning rate of 2 × 10^−4^, which gradually decreases to 1 × 10^−6^. The models were trained for 4000 epochs using the cosine annealing strategy.

To evaluate the performance of our proposed model in dynamic deblurring tasks, we compared our network’s performance with those of MSCNN [5], RNNDeblur [1], SRN [6], and DMPHN [42] using several metrics. Specifically, we used the peak signal-to-noise ratio (PSNR), which quantifies the ratio between the maximum possible signal power and the noise affecting the image—higher PSNR values indicate better image fidelity—and the Structural Similarity Index (SSIM), which assesses the similarity between the restored and ground truth images by evaluating luminance, contrast, and structural features. In addition, we compared parameter size and single-image processing time. For a fair comparison, we assessed the quantitative performance of DSANet2 with data augmentation and DSANet1 without data augmentation. Additionally, we evaluated DSANet++ with the new dataset to demonstrate its validity. The quantitative results are summarized in Table 1. Even DSANet1, which does not use data augmentation, outperformed DMPHN in terms of both metric scores and processing speed. The inclusion of data augmentation improved both the PSNR and SSIM. Furthermore, DSANet++ achieved even better performance with the introduction of the new dataset. These experimental results indicate that the new dataset significantly enhances both quantitative and qualitative performance.

Figure 7 presents the qualitative results obtained using our model. The tested images are blurred images from the test set, without corresponding GT images. The performance of our model was compared with those of MSCNN [5], RNNDeblur [1], SRN [6], and DMPHN [42]. The DSANet performs better in recovering sharp images from complex dynamic scenes compared to previous methods.

We conducted ablation studies on various modules of the proposed network. The DSANet operates at three different scales (*K* = 1, 2, 3), corresponding to input image sizes of 256 × 256, 128 × 128, and 64 × 64, respectively. When *K* = 1, only the original-scale image is input into the network. When *K* = 2, images at two different scales are input. When *K* = 3, images at three scales are processed. Table 2 provides the quantitative evaluation results of the DSANet at different scales, while Figure 8 shows the qualitative evaluation results.

We observe that the performance improvement with a single-scale network is minimal. When *K* = 2, the network achieves good performance but has suboptimal processing speed. The three-scale network brings only slight performance improvements but significantly increases computation speed. These results demonstrate that the multi-scale network is highly effective for image deblurring tasks.

To evaluate the effectiveness of the encoder–decoder structure and the supervised attention module, we designed and compared different models, as follows (outlined in Table 3). 1Ed Model: This network consists of a single encoder and a single decoder, without the supervised attention module. 2Ed Model: This network features two encoders and two decoders but does not include the supervised attention module. 3Ed Model: This network is composed of three encoders and three decoders, excluding the supervised attention module. In contrast, the DSANet represents the final network configuration, which includes three encoders and three decoders, and incorporates the supervised attention module.

In all networks, each encoder and decoder typically consist of an up- or down-sampling layer along with four residual blocks. Adding more residual blocks can expand the receptive field of each scale network. However, stacking too many residual blocks will increase the number of parameters, which can be inefficient. In the DSANet, we used four residual blocks in each encoder and decoder, as adding more than four blocks results in minimal improvement. This design aims to balance efficiency and performance. Additionally, all networks employed the multi-scale architecture introduced by ConvLSTM [43] for a fair comparison.

As the number of encoders and decoders increases, the network’s performance improves due to the larger receptive field. However, this also increases the number of parameters. To balance efficiency and effectiveness, we opted for a network with three encoders and three decoders. The DSANet, with its three encoders, three decoders, and the supervised attention module, demonstrates superior performance. Notably, the DSANet achieves a significant performance boost compared to the 3Ed Model, with only a 0.3 MB increase in parameters. This highlights the effectiveness of the supervised attention module, which enhances feature extraction as a lightweight gating mechanism while reducing redundant features.

Object detection is a major research area in computer vision, with significant advances driven by deep learning methods. However, image blurring remains a challenge that limits object detection performance. To assess the impact of deblurring on object detection, we selected 100 images from the GoPro dataset and evaluated object detection performance before and after deblurring using the Yolov4 algorithm. The qualitative results are shown in Figure 9.

## 6. Conclusions

Images are crucial for information acquisition, but distorted images can hinder this process, making image restoration essential. Image deblurring not only helps users retrieve information more effectively but also enhances performance in tasks such as object detection, segmentation, and classification. However, current deep learning-based deblurring methods face persistent issues. To address these challenges, this paper introduces a new network. The network utilizes a ConvLSTM-based encoder–decoder architecture to accelerate convergence and capture spatio-temporal features. Additionally, we propose a novel supervised attention module to mitigate the high computational costs associated with self-attention mechanisms. This module uses a lightweight gating mechanism to direct the model’s focus towards highly correlated features of the blurred information as it transfers features across networks of varying sizes. To address limitations in traditional image reconstruction evaluation measures, we introduce several loss functions based on the fast Fourier transform. These functions enable the model to learn ambiguity features in the frequency domain effectively. Finally, we have compiled a new dataset that outperforms existing datasets, reducing challenges arising from discrepancies between synthetic and real images. A series of ablation experiments demonstrate the effectiveness of different modules within the DSANet framework.

Beyond its technical contributions, the DSANet offers significant practical implications. The enhanced deblurring performance is particularly valuable in real-world applications where dynamic blurs occur—such as video surveillance, autonomous driving, and robotics—improving not only visual clarity but also the accuracy of subsequent computer vision tasks. By effectively addressing both spatial and temporal variations in blur, the DSANet can contribute to more robust object detection and tracking in safety-critical environments. Moreover, the integration of advanced loss functions and a novel dataset paves the way for future research to further bridge the gap between synthetic and real-world data, making the DSANet a promising candidate for deployment in diverse practical scenarios.

## 7. Discussions

In this study, we proposed the DSANet, which integrates a ConvLSTM-based encoder–decoder architecture with a supervised attention module, to perform dynamic scene deblurring. Experimental results demonstrate that the DSANet achieved excellent performance in both quantitative (PSNR, SSIM) and qualitative evaluations. However, despite the performance improvements, several limitations remain that warrant further investigation.

First, the dataset constructed in this study contributed to improving the model’s generalization ability. However, since it was primarily based on synthetically generated motion blur data, it may not fully capture diverse blur patterns. Therefore, future research should consider utilizing multimodal data to generate more realistic blurred images. Additionally, the model’s performance in complex blur environments was not sufficiently tested in the experiments. This limitation suggests that the model may not fully reflect the diverse blur conditions encountered in real-world scenarios and could lead to an underestimation of its limitations in complex blur environments.

Second, while the PSNR and SSIM were employed to evaluate deblurring performance, these metrics have inherent limitations in fully reflecting perceptual quality. In future studies, we plan to include learning-based metrics such as LPIPS (Learned Perceptual Image Patch Similarity) and user-based evaluations to more accurately analyze the visual quality of the restored images.

Finally, while the integration of spatio-temporal information via ConvLSTM produced meaningful results in dynamic scene restoration, further research on alternative architectures is necessary to effectively handle a wider range of dynamic variations and complex environments.

## Figures and Tables

**Figure 1 sensors-25-01896-f001:**
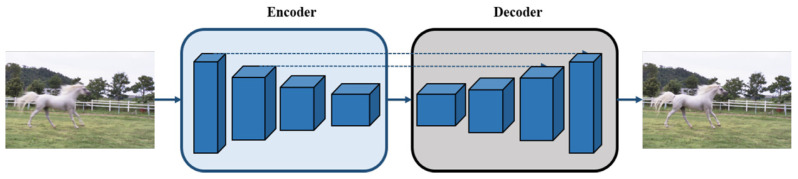
Basic encoder–decoder architecture.

**Figure 2 sensors-25-01896-f002:**
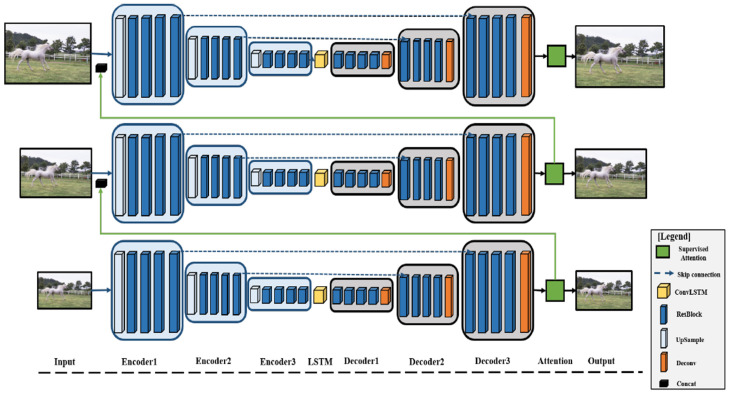
Proposed encoder–decoder architecture.

**Figure 3 sensors-25-01896-f003:**
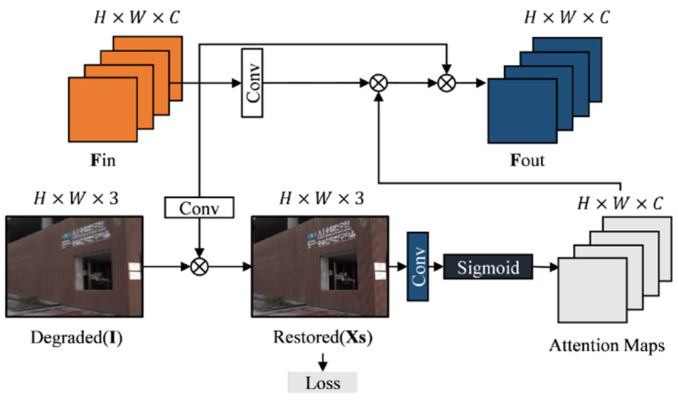
Supervised attention module.

**Figure 4 sensors-25-01896-f004:**
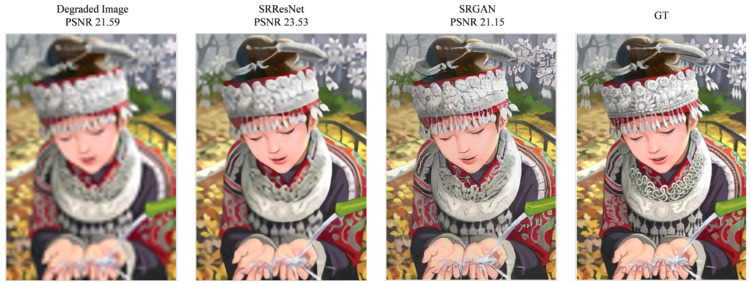
From left to right: degraded image, SRResNet, SRGAN, and GT.

**Figure 5 sensors-25-01896-f005:**
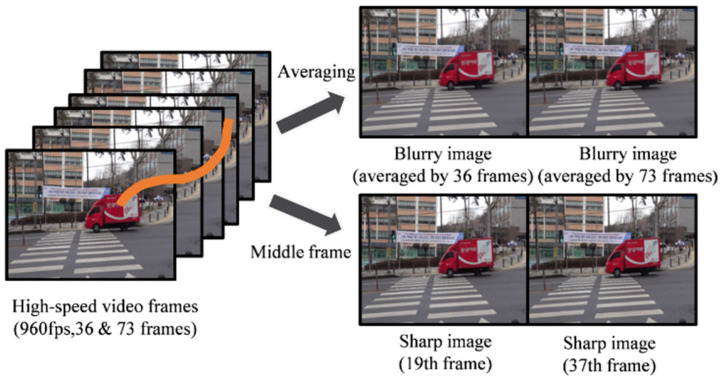
New dataset collection.

**Figure 6 sensors-25-01896-f006:**
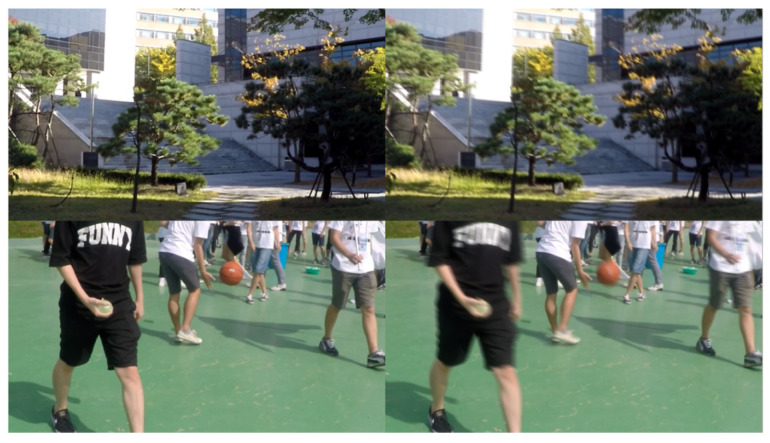
GOPRO dataset pairs of sharp and blur images.

**Figure 7 sensors-25-01896-f007:**
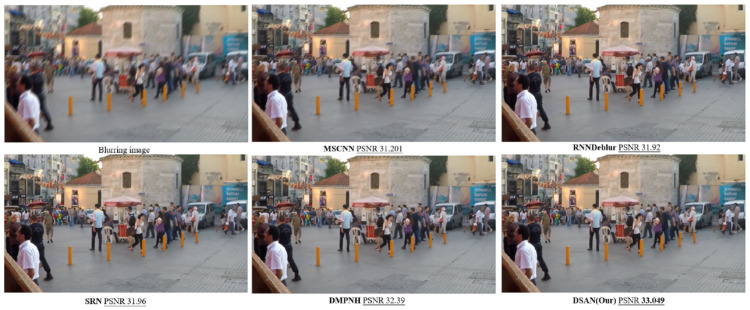
Visual comparison of the benchmark GoPro testing set.

**Figure 8 sensors-25-01896-f008:**
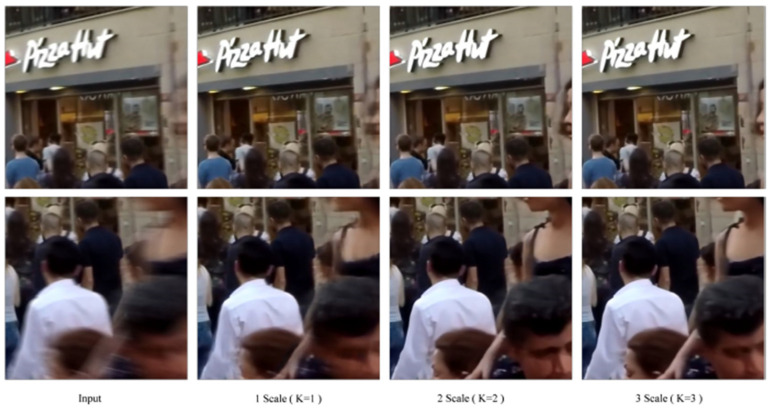
Visual comparison of the multi-scale networks.

**Figure 9 sensors-25-01896-f009:**
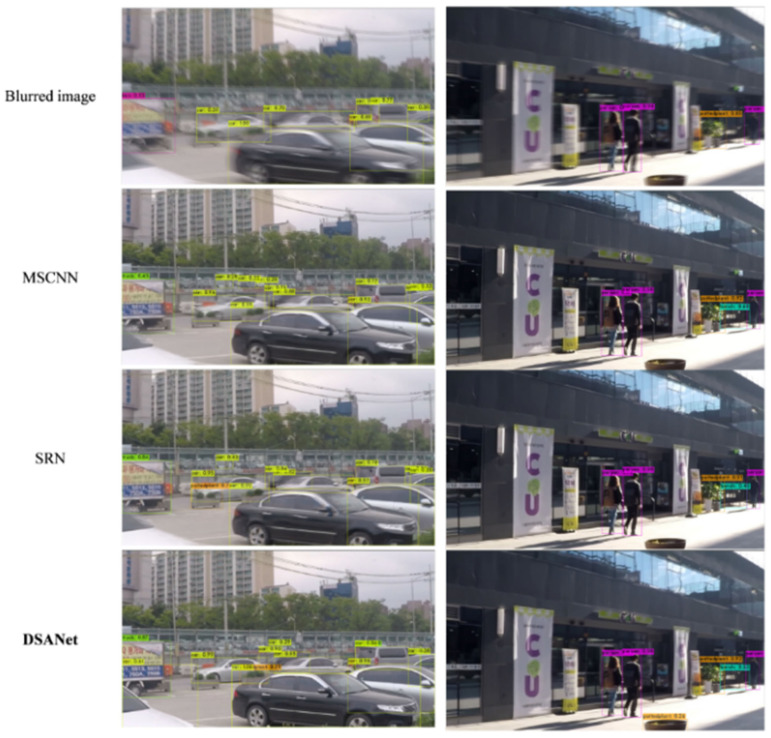
Qualitative evaluation of object detection.

**Table 1 sensors-25-01896-t001:** Evaluation results on the benchmark GOPRO testing set.

Model	PSNR	SSIM	Time (ms)	Size (MB)
MSCNN	29.23	0.9162	4300	303.6
RNNDeblur	29.19	0.9306	1400	37.1
SRN	30.60	0.9323	1600	33.6
DMPHN	31.25	0.9483	424	86.8
DSANet1	31.38	0.9485	254	32.2
DSANet2	31.55	0.9490	254	32.2
DSANet++	**31.65**	**0.9492**	254	32.2

**Table 2 sensors-25-01896-t002:** Quantitative evaluation results of DSANet at different scales.

	*K* = 1	*K* = 2	*K* = 3
PSNR	30.5	31.4	31.65
SSIM	0.9402	0.9485	0.9492
Time (ms)	921	534	253

**Table 3 sensors-25-01896-t003:** Quantitative results for different models.

Model	PSNR	SSIM	Time (ms)	Size (MB)
1Ed Model	27.56	0.9255	80	5.3
2Ed Model	29.4	0.9358	145	12.4
3Ed Model	30.24	0.9401	280	31.5
DSANet	**31.65**	**0.9492**	**254**	32.2

## Data Availability

The dataset presented in this article are not readily available because the data are part of an ongoing study.

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
