# Peer review of "Deep Supervised Attention Network for Dynamic Scene Deblurring"

_sensors, 2025, doi:10.3390/s25061896_

Round 1
Reviewer 1 Report
Comments and Suggestions for Authors
This paper proposes an innovative method based on Deep Supervised Attention Network (DSANet) in the field of dynamic scene deblurring and shows experimental results that are superior to existing methods. However, the paper has shortcomings in the following aspects and needs further improvement: Here are my opinions and suggestions for improvement:
- The introduction does not provide sufficient analysis of existing research and fails to clearly point out the differences between this paper and the literature. It does not clearly explain how DSANet solves the problems existing in current research (such as the redundancy of RNN structure and the high computational overhead of introducing self-attention).
- The literature cited in the full text is outdated or incomplete, and fails to cover the latest research results in this field. It is recommended that the author update the literature review and cite the research results of the past three years to ensure the forward-looking and academic value of the paper.
- Chapter 3 mainly describes the overall architecture of DSANet (such as multi-scale encoder-decoder structure, supervised attention module, and multiple loss functions), while Section 3.3 (lines 272-305) focuses on the construction of a new dataset. The author puts it in parallel with the network architecture, loss function, etc., resulting in unclear logic in the chapter. The dataset section is more suitable as an independent chapter or a sub-chapter of the method chapter.
- (Lines 259-260) The article mentions "We conducted numerous experiments with different loss weights, such as 0.1, 0.4, and 0.6. The results showed that a loss weight of μ=0.1 provided the best performance." The "results" need to be verified.
- In Section 3.3, this study proposed a new synthetic dataset, and the method of making this dataset should be introduced in text. It is recommended that the author describe the steps of data acquisition, blur generation, post-processing, etc. in detail in the dataset chapter, and supplement the statistical comparison with the existing dataset.
- In Section 4 (Lines 320-324), it is mentioned that this paper uses the GOPRO dataset and a self-made dataset to evaluate the performance of the network, but the article does not describe the dataset, such as the resolution of the image.
- (Lines 333-334) In general, the author should introduce PSNR and SSIM. In addition, it is recommended to add the LPIPS (Learned Perceptual Image Patch Similarity) indicator, combine PSNR and SSIM for comprehensive evaluation, and provide qualitative results to verify the perceptual quality improvement of the model.
- The author did not experimentally verify the effect of adding the supervised attention module to the 1Ed or 2Ed model, but only verified it in DSANet (three-scale model). This omission weakens the scientific rigor of the paper. Readers cannot determine whether the supervised attention module is effective at all scales, nor can they quantify its improvement on model performance. Supplement comparative experiments, qualitative results, and ablation experiments to comprehensively evaluate the effect of the supervised attention module and enhance the scientificity and persuasiveness of the paper.
- In lines 393-400, the author uses YOLOv4 to detect objects on deblurred images. The results should be statistically analyzed.
- It is recommended to add a discussion section to analyze the methods and results of this paper and summarize the shortcomings of the research.
- Others:
- The “Where” after the formula does not need to be indented.
- (Lines 159-164): The physical meaning of “hidden state features” is not clearly explained in formula (2). What does “” mean?
Reviewer 2 Report
Comments and Suggestions for Authors
1.Although the article mentions that using the new method to generate datasets can significantly reduce the complexity of synthesized data, it is not explicitly stated whether this method can fully simulate dynamic blur in the real world. If the gap between the synthesized data and the real scene is too large, it may lead to poor performance of the model in practical applications. Multimodal data should be used to assist in generating blurry images that are closer to the real scene.
2.Although the paper compared the performance of the proposed method with some classic deblurring algorithms such as MSCNN, RNNDeblur, SRN, etc., it did not cover the latest deep learning methods such as ESRGAN DMPHN。 The latest deblurring algorithm should be included in the comparative experiment.
3.Although the GOPRO dataset was used for validation in the experiment, the performance of the model on other types of blur, such as non-uniform blur and dynamic blur, was not tested, which may underestimate the limitations of the model in complex fuzzy scenarios. Model performance can be tested on different types of fuzzy datasets.
4. Dynamic Scene Deblurring needs to draw on the latest research trends in machine learning and should at least be mentioned in the analysis, multitasking and multimodal machine learning algorithms. Some machine learning algorithms based on multitasking and multimodality have good reference value for this task, such as the immune depth presentation convolutional neural network used for oil and gas pipeline fault diagnosis, Multi-task learning for hand heat trace time estimation and identity recognition, Deep soft threshold feature separation network for infrared handprint identity recognition and time estimation. Meanwhile, the author should more effectively describe the practical application scope and significance of this article.
Comments on the Quality of English LanguageThe English could be improved to more clearly express the research.
Round 2
Reviewer 1 Report
Comments and Suggestions for Authors
The manuscript has beens sufficiently improved.
With my best wishes.
Reviewer 2 Report
Comments and Suggestions for Authors
The revised version has a very good improvement in algorithm and logic. I warmly recommend publication in present form.